# Oral Fluids for the Early Detection of Classical Swine Fever in Commercial Level Pig Pens

**DOI:** 10.3390/v16030318

**Published:** 2024-02-20

**Authors:** Erin Robert, Kalhari Goonewardene, Ian El Kanoa, Orie Hochman, Charles Nfon, Aruna Ambagala

**Affiliations:** 1Canadian Food Inspection Agency, National Centre for Foreign Animal Disease, Winnipeg, MB R3E 3M4, Canada; erin.robert@inspection.gc.ca (E.R.); kalhari.goonewardene@inspection.gc.ca (K.G.); ian.elkanoa@inspection.gc.ca (I.E.K.); orie.hochman@inspection.gc.ca (O.H.); charles.nfon@inspection.gc.ca (C.N.); 2Department of Medical Microbiology and Infectious Diseases, University of Manitoba, Winnipeg, MB R3E 0W2, Canada; 3Department of Animal Science, University of Manitoba, Winnipeg, MB R3T 2N2, Canada; 4Department of Comparative Biology and Experimental Medicine, Faculty of Veterinary Medicine, University of Calgary, Calgary, AB T2N 4Z6, Canada

**Keywords:** classical swine fever, CSF, oral fluids, aggregate sample, early detection, moderately virulent, highly virulent, swine, diagnostics, surveillance

## Abstract

The early detection of classical swine fever (CSF) remains a key challenge, especially when outbreaks are caused by moderate and low-virulent CSF virus (CSFV) strains. Oral fluid is a reliable and cost-effective sample type that is regularly surveilled for endemic diseases in commercial pig herds in North America. Here, we explored the possibility of utilizing oral fluids for the early detection of CSFV incursions in commercial-size pig pens using two independent experiments. In the first experiment, a seeder pig infected with the moderately-virulent CSFV Pinillos strain was used, and in the second experiment, a seeder pig infected with the highly-virulent CSFV Koslov strain was used. Pen-based oral fluid samples were collected daily and individual samples (whole blood, swabs) every other day. All samples were tested by a CSFV-specific real-time RT-PCR assay. CSFV genomic material was detected in oral fluids on the seventh and fourth day post-introduction of the seeder pig into the pen, in the first and second experiments, respectively. In both experiments, oral fluids tested positive before the contact pigs developed viremia, and with no apparent sick pigs in the pen. These results indicate that pen-based oral fluids are a reliable and convenient sample type for the early detection of CSF, and therefore, can be used to supplement the ongoing CSF surveillance activities in North America.

## 1. Introduction

Classical swine fever (CSF) is a highly contagious hemorrhagic fever in swine, caused by a small, enveloped RNA virus of the Pestivirus genus within the family *Flaviviridae* [1]. Although only one serotype of CSF virus (CSFV) exists, there are five known CSFV genotypes and at least 14 sub-genotypes [2]. Independent of the genotypes, there are three different CSFV pathotypes; low, moderate and highly virulent [3]. Highly-virulent strains of CSFV cause acute disease characterized by high fever, huddling, weakness, anorexia, conjunctivitis, vomiting, constipation followed by watery diarrhea, incoordination, staggering gait, posterior paresis, cough and labored breathing. The skin becomes hyperemic, and develops hemorrhages or a purple cyanotic discoloration. Pigs often die within 1–3 weeks, and convulsions might occur in the terminal stages [4]. The subacute form of CSF is generally caused by moderately-virulent strains. The clinical signs are similar but less severe; the course of infection is prolonged, and the mortality rate is lower. Most recent CSF outbreaks globally are caused by moderately-virulent strains, and the clinical signs are often less severe and atypical [5]. Low-virulent CSFV strains lead to a chronic disease characterized by anorexia, depression, elevated temperature, and periods of constipation and/or diarrhea. Affected pigs usually improve after several weeks but the signs can recur. They show stunted growth, alopecia and skin lesions, suffer from concurrent infections, and die after several months [6].

CSFV is transmitted by direct contact between swine or indirectly via contaminated fomites such as shoes, clothing, and vehicles. The virus can also survive in frozen pork and pork products for years, and therefore, can be transported over long distances and can reappear in countries in which the disease was previously eradicated [7]. Transplacental transmission and perinatal infection of CSFV lead to persistently infected piglets that are seronegative for CSF and shed large amounts of the virus [8,9,10].

Canada successfully eradicated CSF in 1963 and has thus far managed to keep this disease out of the Canadian swine population through the use of strict regulations governing the importation of pigs and pork products. CSF was eradicated from the USA in 1978 and Mexico in 2012 [11,12]. CSF continues to be endemic in Asia, Eastern Europe, Russia, some countries in the Caribbean, and Central and South America. Therefore, North America is at risk of the re-introduction of CSF [13].

The early detection of CSF remains a key challenge, especially when outbreaks are caused by moderate and low-virulent CSFV strains [14]. CSFV is highly contagious compared to its top differential African swine fever virus (ASFV), so it can spread within a pig herd quickly, resulting in higher prevalence prior to the onset of CSF-specific clinical signs [15]. Due to these reasons, enhanced surveillance for early detection is critical for the successful control and eradication of CSF incursions into disease-free areas. Oral fluids are a reliable, cost-effective, non-invasive, aggregate sample type for pathogen surveillance in swine [16]. In North America, oral fluids collected from commercial swine productions are regularly tested for a number of endemic diseases including porcine circovirus type 2 [17], porcine reproductive and respiratory syndrome virus [18], porcine epidemic diarrhea virus [19] and influenza A virus [20].

Using four independent animal experiments, we previously demonstrated that oral fluids can be effectively used to detect both highly-virulent and moderately-virulent ASFV strains in commercial-size pig pens [21]. ASFV genomic material was detected as early as 3–5 days post-introduction of an infected pig and 2–7 days before the infected pig died. Additionally, ASFV genomic material was detected in the oral fluids even after the death of the seeder pig and before contact pigs developed fever and/or viremia.

In the current study, we explored the possibility of utilizing oral fluids for the early detection of CSFV in pigs housed in commercial-size pig pens, infected with highly-virulent or moderately-virulent CSFV strains.

## 2. Materials and Methods

### 2.1. Animals and Animal Housing

In biosafety level 3 animal pens at the National Centre for Foreign Animal Disease (NCFAD), in Winnipeg, Canada, two independent animal experiments were conducted, each with 25 pigs. Experiment #1 was conducted with the moderately-virulent CSFV strain Pinillos [22] and experiment #2 with the highly-virulent CSFV Koslov [23]. The use of animals was approved by the Animal Care Committee at the Canadian Science Centre for Human and Animal Health under the Animal Use Document number C-21-006, and all procedures involving the animals were in compliance with the Canadian Council for Animal Care guidelines.

The animals in both experiments were four to five weeks old, weaned, Large White × Landrace × Duroc cross-bred piglets of mixed sex purchased from the same local commercial supplier in Manitoba. The piglets for each experiment were housed in a single large animal pen which provided 2.4 square feet of floor space per pig, which is slightly higher than the recommended floor space (1.95 square feet per pig) by the Canadian National Farm Animal Care Council Code of Practice for Pigs [24]. The animals were provided with feed twice daily and water ad libitum. The piglets were also provided with a heat lamp as a supplementary heat source and toys for environmental enrichment. After seven days of acclimatization, the pigs were ear-tagged, and marked with spray paint on their backs for easy identification. One of the pigs from each pen was randomly chosen as the ‘seeder pig’, and infected with either CSFV Pinillos or CSFV Koslov as described below.

### 2.2. Virus Propagation, Titration and Inoculation

As the moderately-virulent strain, we used CSFV Pinillos, a genotype 2.6 strain isolated from Colombia in 2016. It is believed to have originated in South-East Asia and was responsible for the CSF outbreaks in Colombia since 2005 [22]. The reference strain CSFV Koslov was used as the highly-virulent strain in our second experiment [23]. CSFV virus propagation and titration were performed according to the NCFAD standard operating procedure (SOP). Briefly, PK-15 cells grown to 50–70% confluence in a T25 cm^2^ rectangular canted neck cell culture flasks with vent cap (Corning^®^, Corning, NY, USA) were inoculated at 0.1 MOI with the respective viral strain. The flasks were then incubated on a rocker for one hour at 37 °C in a 5% CO_2_ incubator. Afterward, the inoculum was removed and 4.5 mL of cell culture media (α-Minimal Essential Medium supplemented with 1% Glutamax, 1% Gentamicin, and 2% irradiated horse serum, all from ThermoFisher, Waltham, MA, USA) was added to each flask. They were then placed in an incubator for three days, frozen at −80 °C overnight, and thawed at 4 °C. The monolayer was scraped and the resulting cell suspension was clarified by centrifugation at 2000× *g* for 20 min. The clarified supernatant was then titered on PK-15 cells to determine the TCID_50_/mL.

On the day of virus inoculation, the seeder pig was removed from the pen to avoid viral contamination within the pen, and inoculated intramuscularly (IM) in the left thigh with 1 × 10^5^ TCID_50_/mL CSFV Pinillos or Koslov while under gaseous anesthesia. The site of inoculation was thoroughly cleaned with 70% ethanol, and the pig was re-introduced to the pen after it fully recovered from the anesthesia.

### 2.3. Clinical Observations

An independent observer entered the animal room each morning to observe the behavior and clinical signs of the pigs. A modified clinical scoring system adapted from Mittelholzer et al. [25] was used to enter a cumulative clinical score for each pig every day. The modifications were made to efficiently capture the visible clinical changes and measurable parameters frequently observed with CSF in group settings, in a simplified and straightforward manner. This includes the addition of neurological signs and rectal temperatures, two important clinical parameters that were not previously taken into consideration. The clinical parameters used for the clinical scoring system are summarized in Table 1. Rectal temperatures were collected during sample collection, under physical restraint or while the pigs were feeding.

### 2.4. Sample Collection

The oral fluid samples were collected as previously described [21], once daily from the pen before feeding the pigs. Briefly, utilizing the TEGO Swine Oral Fluid kit (ITL Biomedical, Reston, VA, USA), two cotton ropes were hung on the front gate of the pen at the shoulder height of the pigs. During the acclimatization period, the pigs were “rope trained” prior to each experiment by leaving the ropes hanging for 60 min, and dusted with a sweet vitamin supplement powder (Prima treats, Bio-Serv, Flemington, NJ, USA) to encourage the pigs to chew on the ropes. This began two days prior to the virus inoculation [−2 days post-infection (dpi)] for CSFV Pinillos and −3 dpi for Koslov, and the sampling continued daily. Please note, dpi and days post-contact (dpc) have the same number; however, dpi references the seeder pig, and dpc refers to the contact pigs. Starting on 0 dpi, ropes were hung for 30 min daily. During rope collection, an outside observer monitored the pigs and recorded the identities of the pigs that chewed on the rope (Appendix A). After a 30-min period, the ropes were removed, the oral fluid was squeezed out of each rope, and pooled into a 50 mL centrifuge tube. The oral fluids were then aliquoted and frozen at −70 °C for later testing.

Individual samples [whole blood (WB), oropharyngeal swabs (OPSW), buccal swabs (BSW), and nasal swabs (NSW)] were collected following the NCFAD standard sampling procedures. In order to collect individual samples, the pigs in each experiment were randomly divided into two sampling groups, and each group was sampled every other day to prevent daily handling and sampling stress. All procedures involving the animals, except for the collection of oral fluids were performed under general anesthesia using isoflurane 2% (5% max) inhalation delivered in 100% oxygen. Pigs were euthanized at their humane endpoints, or at the end of the experiment using sodium pentobarbital (240 mg/mL) injected intravenously. Upon euthanasia, a full post-mortem examination was conducted and several tissue samples were collected.

In experiment #1 (CSFV Pinillos), every other day samples were collected until 12–13 dpi. Thereafter, individual samples were collected on dpi 15–16 and 22–23. Individual samples were also collected if a pig was euthanized as it reached the humane endpoint, or at the end of the experiment (30 dpi). In experiment #2 (CSFV Koslov), every other day samples were collected until 11 dpi and thereafter, when a pig was euthanized as it reached the humane endpoint, or at the end of the experiment (17 dpi). The individual pigs identified for sample collection were removed from the pen, and anesthetized using gaseous anesthesia. Under anesthesia, rectal temperature was measured first, followed by the collection of 3 mL of WB (EDTA) and swabs [sterile Puritan^TM^ cotton swabs (Puritan Medical Products, Guilford, ME, USA) placed in 1.0 mL of Dulbecco’s Phosphate Buffered Saline (Corning Mediatech Inc., Manassas, VA, USA)]. The pigs were returned back to the pen after recovering from anesthesia. All of the samples were aliquoted and frozen at −70 °C until they were tested. Rectal temperatures of the remaining pigs were taken under physical restraint or while they were eating.

### 2.5. Nucleic Acid Extraction and RRT-PCR

Total nucleic acid extraction was performed on WB following NCFAD’s SOPs, utilizing the MagMAX^TM^ Pathogen RNA/DNA Kit (Life Technologies, Burlington, ON, Canada) and the MagMAX^TM^ Express-96 Magnetic Particle Processor (Life Technologies) following the manufacturer’s standard protocol using 55 µL of sample. For the oral fluids, oropharyngeal, buccal, and nasal swabs, total nucleic acids were extracted using the same kit and instrument following the low cell count protocol with 300 µL of sample.

CSFV genomic material was detected following NCFAD’s standard diagnostic real-time RT-PCR assay that targets the 5′ UTR of the CSFV genome [26] with cycling conditions modified to match the fast protocol recommended for the TaqMan^TM^ Fast Virus 1-Step Master Mix (ThermoFisher Scientific, Waltham, MA, USA).

### 2.6. Antibody Detection

For each individual sampling day, a portion of whole blood collected in serum separator tubes (ThermoFisher Scientific) was spun at 3000× *g* for 15 min for clarification before being aliquoted and stored at −20 °C until it was tested. An enzyme-linked immunosorbent assay (ELISA) Test Kit IDEXX CSFV Ab Test (IDEXX Laboratories, Markham, ON, Canada) was used to detect anti-CSFV antibodies in the serum samples. All samples were tested in duplicate, following the manufacturer’s overnight protocol. The absorbance of each well was measured using a SpectraMax microplate reader (Molecular Devices, LLC., San Jose, CA, USA) at 450 nm and the SoftMax Pro software version 7.0 (Molecular Devices). The blocking percentages were calculated according to the manufacturer’s protocol and were considered positive with a blocking percentage ≥ 40%. Samples that had blocking percentages < 30% were considered negative and any sample between the cut-offs was considered suspicious and re-tested. Final serum samples for each pig in both experiments were selected for the initial ELISA. If a pig was suspicious or positive, serially bled serum samples for that pig were tested on the ELISA.

## 3. Results

### 3.1. Experiment #1—CSFV Pinillos

#### 3.1.1. Clinical Picture

The seeder pig developed a mild fever 4 dpi, which fluctuated between a normal temperature to a mild fever throughout the study (Figure 1A). On 8 dpi, it was quiet and slow to rise, but by 13 dpi it was active again. The average rectal temperatures of the pen increased slightly on 13 dpc (Figure 1A), but all pigs including the seeder pig were active, playful, and eating well. On 15 dpc, a few contact pigs were not interested in the rope and were lying down during the oral fluid collection. This behavior continued for several days, and yellowish diarrhea was observed in the pen starting 18 dpc, but the pigs maintained normal appetites. The amount of diarrhea in the pen increased the next day and the seeder pig was noticeably losing weight. Subdermal hemorrhages appeared on the seeder pig on 20 dpi, and its condition continued to deteriorate until 21 dpi when the seeder pig was euthanized as it had reached the humane endpoint (Figure 1A).

Starting 20 dpc, a few contact pigs began shivering and presented with hunched backs. One of the pigs vomited multiple times. On 23 dpc, some of the pigs showed signs of recovery and less diarrhea was observed in the pens. The pigs started eating and drinking, and all pigs chewed on the ropes (Appendix A). As the study progressed, a few pigs developed a cough and exhibited dog sitting posture. The first contact pig was euthanized on 27 dpc as it reached the humane endpoint. On the same day, skin discoloration (red and blotchy) and hemorrhages were observed in several pigs. One of the pigs developed a rectal prolapse, and it was humanely euthanized. On 29 dpc, one pig developed a rash (red) over the entire body, and another pig suffered a 10 min seizure and was subsequently euthanized. All remaining pigs were euthanized on 30 dpc, ending the study.

#### 3.1.2. Detection of CSFV Pinillos Genomic Material

CSFV genomic material was first detected in the seeder pig’s WB on 2 dpi (Ct = 39.09; Figure 2A). In contact pigs, CSFV genomic material was first detected in WB on 11 dpc in one pig (Ct = 38.84; Figure 2A). On 13 dpc, none of the WB samples collected from contact pigs tested positive for CSFV genomic material. However by the next sampling day, i.e., 15 dpc, eleven out of twelve contact pigs tested positive for CSFV genome with the Ct values ranging from 26.35 to 36.05 (Figure 2A). The rest of the contact pigs were tested on 16 dpc and eleven out of twelve tested were positive for CSFV genomic material with the Ct values ranging from 26.77 to 37.97. On the same day, the Ct value of the seeder pig’s WB was 19.26. For the rest of the sampling days, WB from the contact pigs tested positive (Figure 2A).

The OPSW and NSW from the seeder pig were initially detected on 4 dpi, with Ct values 35.14 and 38.43, respectively (Figure 2B,D). The BSW of the seeder pig tested positive starting at 6 dpi (Ct = 34.82; Figure 2C). The Ct values continued to drop for each swab type (due to increasing viral load) and all swab types had their lowest Ct values (OPSW, 25.20; BSW, 25.55; NSW, 20.74) on 12 dpi (Figure 2B–D). Among contact pigs, CSFV genomic material was first detected in an OPSW collected on 8 dpc (Ct = 36.28; Figure 2B). On 9 dpc, CFSV genomic material was detected in NSW collected from two contact pigs (Ct = 39.46 and 39.79; Figure 2D). On 10 dpc CSFV genomic material was detected in BSW collected from three contact pigs (Ct = 36.15, 36.73, and 38.26; Figure 2C). Thereafter, CSFV genome detection in swab samples fluctuated until 22 dpc. After 22 dpc, all three swab types collected from contact pigs tested positive, until the end of the study.

The first CSFV genomic material detection in oral fluid samples was observed on 7 dpi/dpc (Ct = 38.93; Figure 2) and the detection continued until the end of the study. The Ct values for the oral fluid samples continued to drop and the lowest Ct value was observed on 24 dpi (Ct = 21.37).

#### 3.1.3. Detection of Antibodies against CSFV Pinillos Strain

Except for two pigs (pigs 10 and 14), the final serum from all pigs tested negative for anti-CSFV antibodies. Serum from pigs 10 and 14 tested as suspicious. When the remaining serum samples from both pigs were tested, increasing anti-CSFV antibody levels were evident but did not reach the positive cut-off (Appendix A).

### 3.2. Experiment #2—CSFV Koslov

#### 3.2.1. Clinical Picture

During the acclimatization period, soft feces were observed in the pen, and therefore, on −3 dpi/dpc each pig was given 0.5 mL of Tylan 200 (Tylosin 200 mg/mL), a macrolide antibiotic. The condition improved but the feces remained notably soft in the pen until 3 dpi/dpc. The seeder pig developed a high fever on 1 dpi, and thereafter, the rectal temperature fluctuated between high and moderate fever until the pig was euthanized on 13 dpi (Figure 3A). The seeder pig started shivering on 3 dpi and by 5 dpi it was depressed and had a small seizure lasting approximately 10 s.

On 7 dpi, the seeder pig was reluctant to get up and was in the dog sitting position, but it got up and ate when fresh food was offered. On 9 dpi, it was active but then became depressed again towards the end of the day. By 10 dpi the seeder pig developed a rash on the ventral abdomen. In the afternoon, it was given 3.2 mg of METACAM^®^ (20 mg/mL, 0.16 mL) intramuscularly (IM). On dpi 13 the seeder pig developed watery diarrhea and its rectal temperature dropped below normal (hypothermia), and therefore, it was humanely euthanized.

The average rectal temperatures of the pigs in the pen began to rise starting 7 dpc (Figure 3A). By 10 dpc, the majority of contact pigs were slightly depressed and had either moderate or high fevers. On the morning of 11 dpc, there was leftover feed from the previous day and pelleted feces were observed in the pen. About half of the pigs nosed the feed but did not eat. A few contact pigs experienced focal seizures. The next day, only four pigs were eating, and almost all feces were pelleted. One pig developed a high fever (42.2 °C) and it was treated with 3.2 mg of METACAM^®^ (20 mg/mL, 0.16 mL) IM. On 13 dpc, two pigs developed rectal prolapses and five of the contact pigs were euthanized as they had reached the humane endpoint (same day as the seeder pig). The remaining pigs were depressed and mostly ataxic. On 14 dpc pigs remained ataxic, and bleeding from the rectum was noticed in four pigs, and they were later euthanized as they reached the humane endpoint. Later in the day, two pigs developed short seizures lasting less than five seconds. The condition of the pigs continued to decline, and on 15 and 16 dpc, three and five pigs were euthanized, respectively. On 17 dpc, the remaining pigs were euthanized concluding the study.

#### 3.2.2. Detection of CSFV Koslov Genomic Material

CSFV genomic material was first detected in the seeder pig’s WB on 2 dpi (Ct = 28.94; Figure 4A), the first sampling day after inoculation. The seeder pig developed peak viremia on 8 dpi (Ct = 15.88; Figure 4A). CSFV genomic material was detected in one of the contact pigs on 6 dpc (Ct = 38.74). On 8 dpc, the CSFV genome was detected in WB from eight out of twelve contact pigs sampled (Figure 4A). On 9 dpc, WB from nine out of the remaining twelve contact pigs tested positive for CSFV genomic material. Beginning on 10 dpc, WB from all pigs in the pen tested positive for the CSFV genome (Figure 4A).

CSFV genomic material was detected in OPSW collected from the seeder pig starting 2 dpi (Ct = 32.20; Figure 4B) and continued onwards. Both BSW (Ct = 28.18) and NSW (Ct = 25.80) from the seeder pig started to test positive from 4 dpi (Figure 4C,D). The seeder pig continued to test positive for all swab types for the remainder of the study with the highest viral load on 10 dpi for the OPSW (Ct = 19.68) and 13 dpi for the BSW (Ct = 21.01) and NSW (Ct = 19.02).

In contact pigs, CSFV genomic material was first detected in OPSW (Ct = 38.36) on 4 dpc (Figure 4B). Thereafter, the CSFV genome was detected in one NSW (Ct = 37.73), seven BSW (Ct = 37.35–39.01), and eight OPSW (Ct = 37.33–39.06) samples (out of the twelve pigs sampled) on 5 dpc (Figure 4B–D). Starting 8 dpc, the CSFV genome was detected in OPSW samples collected from all pigs, and after 10 dpc, all three swab samples from all contact pigs tested positive for CSFV genomic material.

CSFV genomic material was detected in oral fluid samples starting 4 dpi/dpc (Ct = 39.44), and the amount of genomic material detected in the oral fluids continued to increase until the end of the study (Figure 4).

#### 3.2.3. Detection of Antibodies against CSFV Koslov Strain

Four pigs showed suspicious levels of antibodies in their final serum samples (17 dpc). When the remaining serum samples of these pigs were tested, increasing levels of antibodies were noticed but never reached the positive cut-off (Appendix A).

## 4. Discussion

CSF is a highly contagious, hemorrhagic viral disease of domestic and wild pigs, reportable to the World Organisation for Animal Health (WOAH). It has been eradicated from North America but is endemic in the Caribbean, Central and South America, and Asia. Endemic countries use vaccines to control the disease, and under positive selective pressure resulting from sub-optimal vaccination, CSFV has evolved towards moderately or low-virulent strains [1,22,27]. Such strains do not cause typical signs of the disease for a prolonged period of time, and therefore, can spread within and between herds unnoticed. The introduction of such strains to North America could be devastating as it might not be easily detected based on clinical signs until the disease is widely spread. Therefore, enhanced surveillance efforts in North America for CSF are critical for early detection of a CSF incursion. The currently recommended sample types for CSF surveillance are limited to individual samples such as WB, swabs, semen and serum from live pigs and tissue samples from dead animals [14,28]. Individual sampling of pigs is invasive and costly, and therefore, expanding ongoing surveillance using the currently recommended samples is not practical.

In contrast, oral fluids are an easy-to-collect, aggregate sample type that requires minimal resources and disturbance to the pigs and farm activities. Oral fluids are a mixture of secretions from major and minor salivary glands that combine with oral-mucosal transudate entering the oral cavity by crossing the buccal mucosa from the capillaries [29]. As pigs are constantly sampling their environment by chewing, licking, biting and eating, oral fluid not only represents the health status of the herd but also indicates the contamination of the environment with various pathogens. Atkinson et al. 1993 defined oral fluids as the liquid collected by keeping an absorptive device in the buccal cavity. When it comes to pigs, oral fluid collection is more feasible and fun as it is consistent with pigs’ natural behavior. In fact, being highly social animals, pigs naturally enjoy chewing the ropes, especially when they are in groups [30]. Researchers and practitioners have readily collected oral fluids from groups of pigs over 21 days [31] from barns and abattoirs [32,33], from individually housed animals such as sows [34] and boars [35,36] with training, as well as family oral fluids from litters [37]. Since pigs voluntarily contribute to oral fluid collection, restraint is not necessary, and therefore, it ensures better welfare with less labor and stress to both the animals and the workers. Furthermore, oral fluid collection does not require specific veterinary training and workers in different settings such as barns, holding pens, etc., can be easily trained to collect oral fluids in a straightforward manner, particularly compared to other sample types. It is estimated that the use of oral fluids for pathogen detection can reduce the total number of samples submitted by 23% to 40% during an outbreak [38].

Oral fluids are currently being used in North America for the surveillance of endemic pathogens of swine. Our previous study indicated that oral fluid is a valuable aggregate sample type for the early detection of ASF in pig herds [21]. CSF and ASF are both fatal hemorrhagic fevers in swine and are top differentials in the presence of hemorrhagic signs and lesions. In this study, we show that oral fluids can also be used for reliable and early detection of CSF in a large group of pigs, mimicking commercial pens experiencing an incursion of moderately or highly-virulent CSFV strains. Twenty-five pigs were selected for each study to align with the space available in our BSL3 animal facility, and the resources available for both daily sampling and care of the animals.

As summarized in Table 2, it is important to note that in both experiments, only the seeder pig was viremic at the point of initial CSFV genomic detection in aggregate oral fluids. Although the seeder pig became viremic on 2 dpi with CSFV Pinillos, the initial CSFV genomic detection in oral fluids occurred on 7 dpi/dpc, prior to any pigs within the pen appearing clinically ill (Table 2). When looking at Figure 5, it can be seen that on the same day, CSF was at its lowest prevalence in the pen (i.e., 4%). CSFV Pinillos infected almost all contact pigs by 16 dpc, but the average clinical score in the pen remained low for several days (21 dpc). This particularly highlights the ability of moderately-virulent CSFV strains to spread in naïve pig herds rapidly but go unnoticed for several weeks.

As expected, CSFV Koslov induced high fever and severe clinical signs in the seeder pig, and the disease spread through the pen faster in comparison to that observed with CSFV Pinillos. As a result, experiment #2 was terminated earlier (17 dpi) than the first experiment (30 dpi). The seeder pig inoculated with CSFV Koslov developed fever on 1 dpi and viremia on 2 dpi (Ct = 28.94), the first day of sampling after infection (Table 2). It is likely that the seeder pig developed viremia on 1 dpi, coinciding with the fever, but no sampling was conducted on that date to support this argument. CSFV genomic material in oral fluid samples was first detected on 4 dpi, with CSF prevalence, again, at 4% within the pen (Figure 5).

Within both experiments, there were particular days when the seeder pig did not chew on the ropes to contribute to the oral fluid sample but CSFV genomic material was detected in the oral fluids (Appendix A). This phenomenon was observed on 9 dpi/dpc and 5–6 dpi/dpc for CSFV Pinillos and Koslov, respectively. On these days in their respective experiments, the contact pigs were not yet viremic but some individual swab samples collected show detectable levels of CSFV genomic material (Figure 2 and Figure 4 and Table 2). A similar observation was made in our previous ASF study [21]. Similar to this study, it is highly likely that the virus is shed by the seeder pig, picked up in the environment by contact pigs, and deposited on the rope during the 30 min oral fluids collection period.

The amount of oral fluids collected each day during both experiments varied from 2 to 28 mL and the volume was in line with the number of pigs that chewed on the rope on a given day (Appendix A). Additionally, the volume of oral fluids collected was an indirect indication of the overall health of the pigs in the pen. As the infection progressed, the number of pigs interested in the ropes decreased, and therefore, the volume of oral fluids collected decreased. Despite the decreasing volume and fewer pigs chewing the ropes, the amount of CSFV genome detected in the oral fluids increased over time.

Overall, these results clearly show that pen-based oral fluid samples can be used for early detection of CSF incursions in commercial-size pig pens. CSFV was detected by 4 dpi/dpc with the highly-virulent CSFV strain Koslov and by 7 dpi/dpc with the moderately-virulent strain CSFV Pinillos. The difference in detection time could be attributed to the extent of replication of each virus in the infected pig. CSFV Koslov replicated faster and reached higher titers in the blood of the seeder and contact pigs, in comparison to that of the CSFV Pinillos strain (Figure 2A and Figure 4A).

In both experiments, CSFV was detected in the oral fluid when there was only one pig infected, and with the absence of clinical signs indicating the presence of CSF in the pen. Despite the mild fever, the seeder pigs were playful and eating normally. It is also noteworthy that with both the highly and moderately-virulent CSFV, oral fluid detection occurred prior to any of the contact pigs becoming infected. The ability to detect CSFV in commercial pig pens before the virus spreads to contact animals minimizes the viral load in the affected farms, and therefore, significantly reduces the possibility of spreading the virus to nearby pens/farms. A reduced viral load on the farm and/or within the pen also favors the conditions for successful disinfection of the affected premises. The ability to detect CSF early before it spreads to contact animals is also important if countries decide to replace a total stamping-out policy with a test-and-removal policy to minimize the economic burden on farmers or to alleviate animal welfare concerns related to mass killing.

In line with our previous observations with ASF [21], once the oral fluids tested positive for CSFV genomic material, they continued to test positive until the end of the experiment. However, unlike in the ASF experiment, CSFV-infected seeder pigs lived longer (21 days with CSFV Pinillos and 13 days with CSFV Koslov) and some of the contact pigs became viremic before the seeder pigs reached their humane endpoints. As a result, the contact pigs in both experiments described in the current study, directly contributed to the CSFV genome detection in oral fluid samples even prior to the seeder pig’s death.

In both CSF experiments reported here, the CSFV genome was detected earlier and stronger in individual sample types collected from the seeder pigs than in pen-based oral fluid samples. However, due to the lack of obvious clinical signs, the identification of infected pigs during the early stages of a CSF incursion is not possible. Therefore, targeted individual blood or swab samples from infected pigs in commercial settings for routine surveillance of CSF is unlikely to be useful for early detection. Random sampling of a portion of pigs in large farms is feasible but is less reliable than using oral fluid for early detection. As shown in Figure 5, the percentage of positive detection for CSF in individually collected samples remained less than 100% until much later (20 dpi with CSFV Pinillos and 9 dpi with CSFV Koslov) in the CSF infection. As a result, incorporating frequent oral fluid collection and testing into the ongoing surveillance efforts will better facilitate the early detection of CSF rather than random individual sampling. Among individual samples for the early detection of CSFV genomic material, WB was the best followed by OPSW. CSFV genomic material was detected later in buccal and nasal swabs.

The use of oral fluids for early detection of CSF has been previously studied. However, to the best of our knowledge, no other group had attempted to use oral fluids as an aggregate sample type for the early detection of CSF in a large group with a single pig infected. Those studies used either small groups of pigs, collected oral fluids from individual pigs, or simultaneously inoculated all of the pigs with CSFV. The results from those studies, however, are generally in agreement with the results obtained in this study. In 2017, Dietze et al. reported the detection of CSFV genomic material as early as 7 dpi in a ”rope-in-a-bait” based oral fluid samples collected from pigs that were simultaneously inoculated IM, with the moderately-virulent genotype 2.3 strain CSFV Alfort/Tubingen [39]. In line with these findings, also using individual oral fluid samples obtained using 25 cm long cotton ropes from pigs infected intra-nasally with the highly-virulent strain CSFV Alfort/187 (genotype 1.1), Petrini et. al. showed CSFV genome detection around 8 dpi [40]. In a similar study, Panyasing et al. demonstrated the detection of CSFV genomic material in oral fluids from pigs intra-nasally infected with the highly-virulent strain ALD, as early as 4 dpi. In the study, 30 pigs were used and they were individually housed. Cotton ropes hung in the pen for 30 min prior to their morning feeding were used to collect oral fluids from individual animals [41]. In domestic pigs inoculated with a low-virulent Haiti-96 strain, CSFV was detected in oral fluids as early as 5 dpi [42]. In the current study, seeder pigs were inoculated via the IM route rather than the oro-nasal route, to minimize possible contamination of the environment by the inoculum. However, it is possible that if the seeder pigs were infected naturally via the oro-nasal route, CSFV genomic material could have been detected even earlier than 4 dpi for CSFV Koslov and 7 dpi for CSFV Pinillos.

Oral fluids have also been evaluated previously for the detection of antibodies to CSFV [43]. Since none of the pigs in this study seroconverted, no attempt was made to detect anti-CSFV antibodies in the oral fluid samples collected. Generally, in CSF-infected pigs, antibody detection can take 2–3 weeks [44]. In this study, it is important to remember that the number of days post-introduction of the inoculated seeder pig is not equivalent to the number of days post-infection for the contact pigs. The experiment using CSFV Koslov was terminated on 17 dpi. When testing the serial bleed, pig #20 was the only one to reach suspicious levels of anti-CSFV antibodies, and this pig developed viremia on 8 dpc (9 days of viremia). While the experiment with the moderately-virulent strain, CSFV Pinillos, continued for 30 days, only two pigs (pigs #10 and 14) showed suspected levels of anti-CSFV antibodies in their final serum collection (Appendix A). They developed viremia on 15 and 16 dpc, indicating they were only around 2 weeks post-infection when the study ended. During a previous pathogenesis study [22] using the same strain, only one out of six pigs that were oronasally inoculated had seroconverted by 22 dpi.

Oral fluid often contains environmental contaminants including fecal matter and feed, which could potentially have inhibitory effects on the performance of downstream diagnostic assays such as PCR [45]. Furthermore, the amount of viral genome present in oral fluid is low, especially when there is only one infected pig present within the pen. Therefore, when oral fluid is used in surveillance for the early detection of pathogens including CSFV, the laboratories must use highly efficient nucleic acid extraction methods available such as magnetic bead-based extraction methods, and highly sensitive RT-qPCR assays for CSFV genome detection.

This study further proved again that pigs housed in groups would chew the ropes until they reached their humane endpoints, ensuring the possibility of obtaining a reliable sample containing pathogens circulating in a pen. In the ongoing efforts to protect North American swine herds against CSF and ASF, we recommend the incorporation of oral fluid testing into the ongoing surveillance methods for commercial swine herds. To minimize the laboratory costs, a multiplex real-time PCR assay targeting ASFV and CSFV can be validated and used [42]. In the event of a positive or suspicious level of CSF or ASF detection in aggregate oral fluid samples, the source farm can be traced back and individual samples can be collected and tested for confirmation.

## 5. Conclusions

This study has shown that rope-based oral fluids can be used as a feasible aggregate sample type for early detection of CSFV in commercial-level pig pens. This is especially true for the early detection of moderately-virulent CSFV, the most prevalent circulating CSFV strains in endemic countries. These strains typically show mild and non-specific clinical signs for weeks, and therefore, can be easily missed. Oral fluids, when combined with other surveillance tools such as screening for suspected diseased and dead pigs, should strengthen the current toolbox for the early detection of devastating transboundary swine diseases such as CSF and ASF.

## Figures and Tables

**Figure 1 viruses-16-00318-f001:**
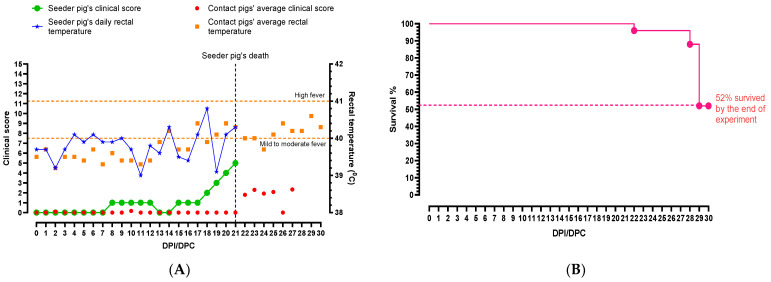
Summary of (**A**) clinical parameters and (**B**) survival percentages of pigs infected with CSFV Pinillos. The seeder pig’s rectal temperature started increasing after 4 dpi, and fluctuated throughout the study. The highest rectal temperature (40.8 °C) of the seeder pig was observed on 18 dpi. The average rectal temperature of the pen started rising on 14 dpi, and fluctuated until the end of the study. The seeder pig was euthanized on 21 dpi as it reached the humane endpoint. The clinical score of the pen remained low until the seeder pig was euthanized on 21 dpi, and slightly increased after. Average rectal temperatures of contact animals are presented for clarity. By the end of the experiment, 52% of the animals survived and some showed signs of recovery.

**Figure 2 viruses-16-00318-f002:**
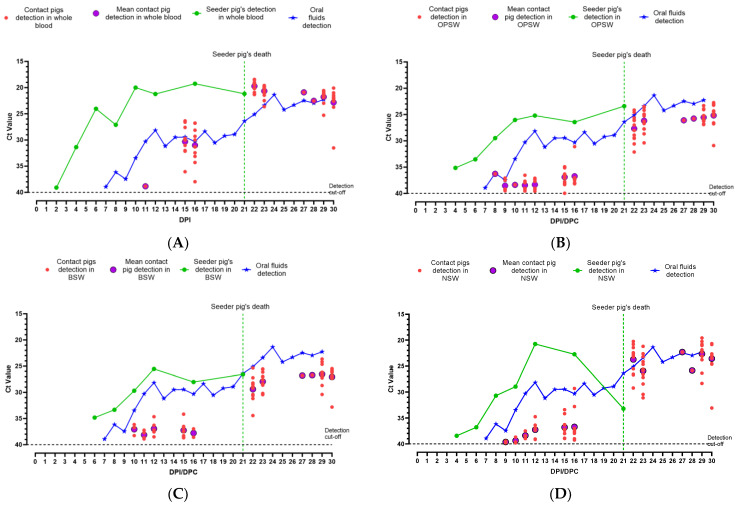
Summary of detection of CSFV genomic material in clinical samples collected from pigs inoculated with CSFV Pinillos: (**A**) whole blood (WB); (**B**) oropharyngeal swabs (OPSW); (**C**) buccal swabs (BSW); (**D**) nasal swabs (NSW) in comparison with oral fluid detection.

**Figure 3 viruses-16-00318-f003:**
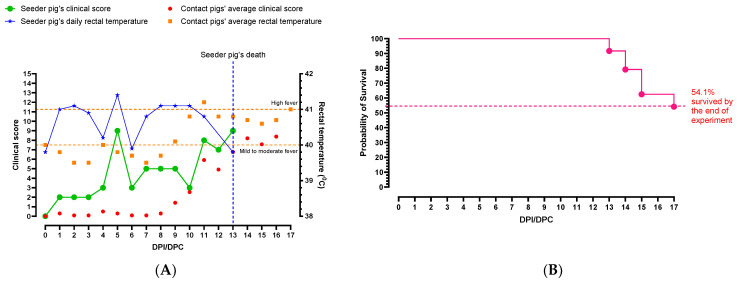
Summary of (**A**) clinical parameters and (**B**) survival percentages of pigs infected with CSFV Koslov. The seeder pig’s rectal temperature started increasing on 1 dpi and fluctuated throughout the study. The highest rectal temperature of the seeder pig was observed on 5 dpi (41.4 °C). The average rectal temperature of the contact pigs started rising on 7 dpc, continued to rise until 11 dpc and stayed high until the end of the study. The seeder pig was euthanized on 13 dpi as it reached the humane endpoint. The average clinical score of the pen remained low until 8 dpc then continued to increase after. Average rectal temperatures of contact animals rather than individual animal temperatures are shown for clarity. Only 54% of the animals survived to the end of the experiment (17 dpi).

**Figure 4 viruses-16-00318-f004:**
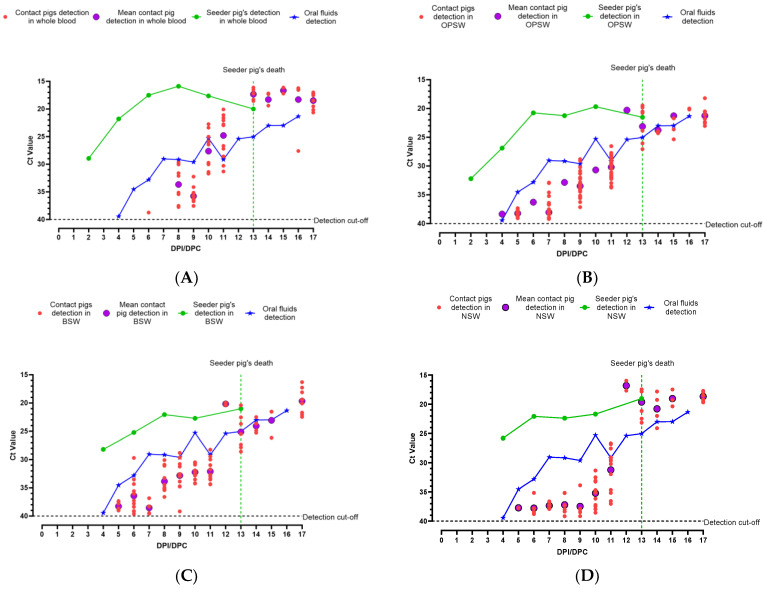
Summary of detection of CSFV genomic material in clinical samples collected from pigs inoculated with CSFV Koslov: (**A**) whole blood (WB); (**B**) oropharyngeal swabs (OPSW); (**C**) buccal swabs (BSW); (**D**) nasal swabs (NSW) in comparison with oral fluid detection.

**Figure 5 viruses-16-00318-f005:**
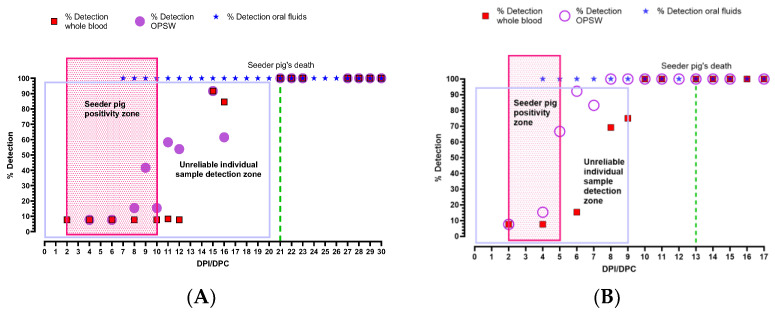
Estimation of percentage detection of CSFV genomic material in aggregate oral fluid samples and individual whole blood samples. (**A**) CSFV Pinillos; (**B**) CSFV Koslov. After its first detection in oral fluids, CSFV genomic data remained at 100% detection throughout the study. Percent detection of CSFV genomic material in individual blood and oropharyngeal swab samples remained lower than 100% for many days afterward (Unreliable individual sample detection zone). Seeder pig positive zone = the period when only the seeder pig was viremic.

**Table 1 viruses-16-00318-t001:** Clinical scoring parameters used for the CSF experiments. Scores from each parameter were added to a maximum cumulative clinical score of 24.

	Parameter	Criteria	Score
1	Rectal temperature	38–40 °C (Normal)	0
		>40 °C but < 41 °C (mild to moderate fever)	1
		≥ 41 °C (High fever)	2
		Temperature < 38 °C (Hypothermia)	3
2	Behavior & mentation	Normal, alert, responsive	0
		Slightly reduced liveliness, stands up unassisted, resists restraint & rectal thermometer	1
		Obtunded, tired, reluctant to get up unassisted, lies down quickly, reduced resistance to restraint & rectal thermometer	2
		Stationary, moribund, unconscious, non-responsive	3
3	Walking	Normal, coordinated	0
		Slow, hesitant, crossed legged	1
		Distinct ataxia, lameness	2
		Severe lameness, unable to walk	3
4	Skin	Normal, evenly pink, hair coat flat	0
		Reddened skin areas	1
		Purple discolored & cold skin areas, few areas of petechiae	2
		Large areas of black-red discoloration, no sensitivity, large hemorrhages on skin	3
5	Appetite	Greedy, hungry, voracious	0
		Eats slowly	1
		Sniffs food but not eating	2
		No interest in food at all	3
6	Gastrointestinal	Normal, soft feces	0
		Reduced amount of feces, dry or pelleted	1
		Diarrhea or small amount of blood in feces or melena or fibrin-covered dry feces	2
		Severe watery diarrhea, bloody diarrhea or no feces or mucus in rectum	3
7	Respiratory	Normal breathing	0
		Increased respiratory rate, dyspnea	1
		Coughing, loud respiratory sounds, rales	2
		Severe dyspnea, open mouth breathing	3
8	Neurological signs	Normal	0
		Dog sitting	2
		Unambiguous neurological signs (seizures, convulsions)	3
Maximum Cumulative Clinical Score			24

**Table 2 viruses-16-00318-t002:** Summary on pig mortality and initial CSFV genome detection in oral fluids, whole blood, and oropharyngeal swabs. Please note that the mortality includes pigs that were euthanized as they reached humane endpoints. Values in parenthesis are Ct values.

Exp.	Number of Pigs	CSFV Strain	Initial Detection in OF	Initial Detection—Seeder Pig		Initial Detection—Contact Pigs		Seeder Pig Mortality	Herd Mortality at the Study End
				WB	OPSW	WB	OPSW		
1	25	Pinillos	7 dpi (38.93)	2 dpi (39.09)	4 dpi (35.14)	11 dpc (38.84)	8 dpc (36.28)	21 dpi	48%
2	25	Koslov	4 dpi (39.44)	2 dpi (28.94)	2 dpi (32.20)	6 dpc (38.74)	4 dpc (38.36)	13 dpi	45.90%

## Data Availability

The authors will provide data related to this study upon request.

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
