# Peer review of "Oral Fluids for the Early Detection of Classical Swine Fever in Commercial Level Pig Pens"

_viruses, 2024, doi:10.3390/v16030318_

Round 1
Reviewer 1 Report
Comments and Suggestions for Authors
Comments and Suggestions for Authors
The manuscript “Oral Fluids for the Early Detection of Classical Swine Fever in 2 Commercial Level Pig Pens” is an interesting and insightful study on the use of oral fluid in effective and reliable CSFV detection in pig pens. The paper may contribute to the literature by giving important information to support oral fluid as a new, easier, and more convenient sample type useful for diagnostic approaches in CSF surveillance programs. The manuscript is properly organized, well written and provides tables and images to better understanding of the results.
I believe this manuscript could be a valuable publication, but it needs some revision before publication.
MAJOR COMMENTS
1. On line 124: The Authors reported a “slightly modified clinical scoring system”. The clinical scoring suggested by Mittelholzer et al (2000) represents the standard checklist for the determination of the clinical score in CSF animal experiments. A comparison between the two checklists shows several important differences (not slight). Some parameters were removed (i.e., body tension, body shape, eyes/conjunctiva, and leftovers in feeding through), others were added (i.e., rectal temperature, neurological signs), and further others were modified. Consequently, the maximum cumulative clinical score differs between the two scoring systems. When a standard method is available, it would be desirable always to use that to allow a simple and clear comparison among studies. On the other hand, I understand that some changes carried out by the Authors allow to detect important clinical signs (for example, rectal temperature) which were not previously taken into consideration. I suggest explaining better the changes made to the checklist in the M&M and giving the reasons in the discussion. The modified version might be suggested as an alternative new standard reference for clinical scoring when the rectal temperature can be delivered.
2. The Authors reported several times in the text the results obtained in another previously published study on ASFV (Goonewardene et al., 2021, Evaluation of oral fluid as an aggregate sample for early detection of African swine fever virus using four independent pen-based experimental studies). Although the comparison between the two studies is pertinent and useful to catch the novelty of the work (oral fluid of pig pen as an aggregate sample vs individual samples; pigs infected by seeder vs pigs experimentally infected by IM virus inoculation), lines 373-382 appear not necessary. I suggest removing them.
MINOR COMMENTS
On line 35: remove the comma after (CSF)
On line 151-152: “Individual samples (whole blood; WB, oropharyngeal swabs; OPSW, buccal swabs; BSW, and nasal swabs; NSW)” should be modified in “Individual samples (whole blood (WB), oropharyngeal swabs (OPSW), buccal swabs (BSW), and nasal swabs (NSW))”.
On line 396: Insert reference 21 after “fluid study.”
Reviewer 2 Report
Comments and Suggestions for Authors
This is an interesting and useful study. and the manuscript is generally well-written. I have two concerns that need to be addressed as indicated below. Some edits required are listed.
In the discussion, it is necessary to analyse your findings and compare them between your two experiments and to existing studies if there are any. The discussion is good up to line 282 and from Line 488 onwards. However, the two paragraphs from Lines 383-429 contain too much detail that duplicates the results. One paragraph comparing the overall results from the two studies should suffice, indicating a shorter progression and higher viral loads with the highly virulent strain, and emphasizing the oral fluid results but preferably without repeating them. Try to summarize the findings without going into detail again about the progression of the clinical signs and test results, all of which have already been fully described and illustrated. Assume that the reader did read the results section and can go back to it if necessary. Table 2 and Figure 5 in my opinion belong in the results section, where they will be helpful to the reader as a summary of the rather complicated graphics in figures 1-4.
It seems curious that none of the pigs sero-converted. Antibodies induced by CSF infection can usually be detected 11-14 days post-infection, and the absence of antibodies is usually suggestive of failure of infection, yet the pigs were viraemic and did show clinical signs. I think that this deserves further discussion. The way it is written, it seems that the decision not to test oral fluid for antibodies was taken because none of the pigs-seroconverted, but antibodies are not mentioned anywhere else in the document, so how do you know that none of the pigs sero-converted? Please add some explanation of how this was determined, and also provide some suggestions as to why this should have been so and if there are precedents for this. Determination of freedom from CSF depends largely on serology so it is disturbing to know that the disease may be present in spite of the absence of antibodies for up to 30 days post-infection!
Suggested edits:
Line 46: Insert a comma before however, or preferably replace however with but: ‘similar but less severe’ is better English.
Line 116: Replace ‘tittered’ with ‘titered’ – ‘tittered’ means ‘laughed nervously’!
Line 126: Replace ‘are summarized’ with ‘is summarized’, or omit ‘The checklist’, which currently is the subject of the sentence so should take a singular verb.
Lines 140 and 146: 60 minutes and 30 minutes should not be hyphenated. These should only be hyphenated when used as an adjective, as in Line 148.
Lines 151-152: Use either a comma or a colon between the name of the sample and the acronym, the use of a semi-colon here is incorrect.
Line 194: Replace ‘laying’ with ‘lying’, the former is widely used in speech but remains incorrect, as it refers to laying something down, or laying an egg, but definitely should not be used for adopting a prone position, it really is very bad English.
Line 195: Insert a comma after however – a comma should be placed before and after however when it used instead of but (usually but would be preferable).
Line 338: Replace Organization with Organisation. US spell checkers automatically change the name, and one sees it increasingly frequently, but the official name of a body like WOAH should not be changed by a spell checker, the choice of the name does not rest with computer software!
Lines 381-382: I suggest adding ‘or mortality’ after clinical signs, because commercial pigs are usually not observed as carefully as pigs in an experiment, and because of the slow spread of ASF it can take some time for the mortality to exceed levels considered normal.
Lines 385-386: ‘When CSFV was inoculated…’ would be easier to understand (but as indicated above, a complete rewrite of this and the next paragraph is needed).
Comments on the Quality of English Language
The English is generally good, there are a few peculiarities and errors that are listed in my comments on the manuscript.
Reviewer 3 Report
Comments and Suggestions for Authors
This paper describes that the gene detection by real-time RT-PCR from oral fluids using cotton rope is suitable for early detection of classical swine fever virus (CSFV) infection in a swine herd. The experimental design was reasonable, and the interpretation of the data was logical and relevant. The reviewer requests the following additional information to improve the quality of the paper.
Major comments
1. L25-26, Abstract: It should be clearly stated that viral genes detected in oral fluids in the early stages of infection are considered to be derived from excretions into the environment from the Seeder pig.
2. L336, Discussion: Does this method prove to be useful not only for fattening pigs but also for breeding pigs? The usefulness and feasibility of this method for breeding pig farms should be discussed.
3. L336, Discussion: An additional comment is needed regarding whether the study design of 24 naïve cohabiting pigs per seeder pig (total 25 pigs) reflects the standard housing density for each pen of a pig farm in North America.
4. L336, Discussion: This reviewer understands and respects that North America follows a non-vaccination policy on pig farms against CSFV infection. However, this disease is still controlled by vaccination in other regions. Could the virus be detected in oral fluids in the vaccinated farm if naïve pig due to the vaccine failure was infected with CSFV as the Seeder pig? Considering that the virus detected in the oral fluids on day 4 (highly-virulent strain) or day 7 (moderately-virulent strain) is derived from the Seeder pig, could the virus be detected on the same period from oral fluids even if the cohabiting pigs were not immunologically naïve to CSFV? Please discuss this point.
5. L383, Discussion: The standard method for virus detection is the measurement of the infectivity titer (TCID50 method) using pig-derived cells. Indicate the infectivity titers of day 4 (highly-virulent strain) and day 7 (moderately-virulent strain) samples when the virus was first detected in oral fluids.
Minor comments
1. L94: Add brief information about the size of the animal pen for 25 pigs to understand your experimental design.
2. L122: Indicate the humanistic endpoint scores (numbers) clearly for euthanasia.
3. L519–521: The detection limit of your real-time RT-PCR should be indicated as the copy number of the viral genomes.
4. L557 Reference: There are several references cited where all the initial letters of the words are capitalized in the title of each article. In addition, some reference papers contain no information about the journal title.
